# An Investigation of Laser Produced Lead-Tin Alloy Plasmas between 10 and 18 nm

**Enda Scally [1],\*, Fergal O'Reilly [2] , Patrick Hayden [2], Isaac Tobin [2] and Gerry O'Sullivan [2]**

[1]   School of Electronic and Electrical Engineering, Technological University Dublin, D08 NF82 Dublin 8, Ireland
[2]   School of Physics, University College Dublin, Belfield, D04 N2E5 Dublin 4, Ireland; f.oreilly@ucd.ie (F.O.);
      patrick.hayden@ucd.ie (P.H.); todindi@tcd.ie (I.T.); gerry.osullivan@ucd.ie (G.O.)
\*   Correspondence: enda.scally@tudublin.ie

**Abstract:** The results of a systematic study performed on Pb-Sn alloys of concentration 65–35% and 94–6% by weight along with spectra from pure Pb and Sn in the wavelength range of 9.8–18 nm are presented. The dynamics of the Nd:YAG laser produced plasma were changed by varying the focused spot size and input energy of the laser pulse; the laser irradiance at the target varied from $7.3 \times 10^9$ W cm$^{-2}$ to $1.2 \times 10^{12}$ W cm$^{-2}$. The contributing ion stages and line emission are identified using the steady state collisional radiative model of Colombant and Tonon, and the Cowan suite of atomic structure codes. The Sn spectrum was dominated in each case by the well-known unresolved transition array (UTA) near 13.5 nm. However, a surprising result was the lack of any enhancement or narrowing of this feature at low concentrations of Sn in the alloy spectra whose emission was essentially dominated by Pb ions.

**Keywords:** Pb-Sn alloys; EUV emission of high Z materials; collisional radiative model; Cowan suite of Codes

## 1. Introduction

The emission from Sn in a 2% bandwidth at 13.5 nm arising from $\Delta n = 0$, n = 4–4 transitions, has become the definitive source of choice for extreme ultraviolet (EUV) lithography systems. This emission results from an intense unresolved transition array (UTA) whose width and intensity are strongly influenced by plasma opacity [1]. For pure Sn targets, plasmas produced by a solid state λ~1 μm laser, the UTA is broad and overlaid with numerous strong absorption lines, while with decreasing concentration the intensity increases while the width decreases significantly due to increased radiation transport of the most intense transitions and the intensity reaches a maximum at a concentration of a few % [2]. For plasmas produced with $CO_2$ lasers (λ = 10.6 μm) the reduction in critical electron density (and consequently ion density) results in a greater UTA intensity and laser to EUV emission conversion efficiency (CE); again due to the reduction in plasma opacity resulting from a lower ion density [3]. Consequently, $CO_2$ lasers are used to excite the UTA emission in current lithography sources. Despite the advances in EUV lithography, work is still ongoing regarding metrology sources for reflectometry, surface patterning, surface chemical analysis, etc. [4]. Fahy *et al.* proposed a liquid collector optic to mitigate the debris associated with laser produced plasmas (LPPs) and multilayer optics [5]. Following this Kambali *et al.* proposed using LPPs from Au or an Au-Sn alloy as a metrology source in the 10 to 18 nm region where Au has a relatively flat spectrum with the added attraction that the alloy had a melting temperature of 280 °C and thus could be used as an easily renewable plasma fuel [6]. Similar work by Tobin *et al.* in the 10 to 18 nm region utilised a room temperature gallium alloy (galinstan) source in a laser triggered discharge plasma (LDP), they obtained a relatively flatter

spectrum with respect to a pure Sn plasma and an emission structure that reflected the presence of its composite elements [7].

Here the results of a systematic study performed on Pb-Sn alloys of concentration 65–35% and 94–6% by weight, along with spectra from pure Pb and Sn, in the wavelength range of 9.8–18 nm is presented. The dynamics of the Nd:YAG laser plasma were changed by varying the focused spot size and input energy of the laser pulse. The contributing transitions and thus ion stages were identified using the Hartree Fock with configuration interaction (CI) suite of codes written by Cowan [8,9]. To aid in understanding the binary plasmas, the results are compared to the optically thinner $CO_2$ plasmas and thus contribute to building a database of the spectral emission of high Z materials in the 9.8–18 nm region. The Pb-Sn alloy which in a ratio of 65–35 has a minimum eutectic melting point of $\approx$ 183 °C, has the potential to provide a clean and bright broadband microscopy or reflectometry EUV source when combined with a suitable renewable liquid optic.

## 2. Experimental

A schematic of the experimental set-up is shown in Figure 1, where; 1: is a Continuum Surelite III Nd:YAG laser system with a full width at half maximum intensity (FWHM) pulse of 5 ns, and an $M^2 = 3.5$; 2: half wave-plate + beam-splitter energy selector for the Nd:YAG laser; 3: labels the Optosystems Infralight SP10 gas flow transversely excited atmospheric (TEA) pulsed laser system ($CO_2$ laser; $M^2 = 4$); 4: the $CO_2$ pulse plasma shutter; 5: the steering optic; 6: the $CO_2$ pulse profiler; 7: the $CO_2$ laser focusing lens (f = 20 cm) and 8: the Nd:YAG laser focusing lens (f = 10 cm). Both the lens and target can be moved in vacuum using an interface program to control the actuators to a precision of 1 μm. As seen from Figure 1, the Nd:YAG laser was incident along the target normal, the $CO_2$ laser was incident at 45 degrees to the target normal and the spectrometer viewed the plasma at 45 degrees to the target normal. The focal position of the lens was found by finding the local minimum value in CE for a 2% bandwidth at 13.5 nm as a function of lens position close to focus [10].

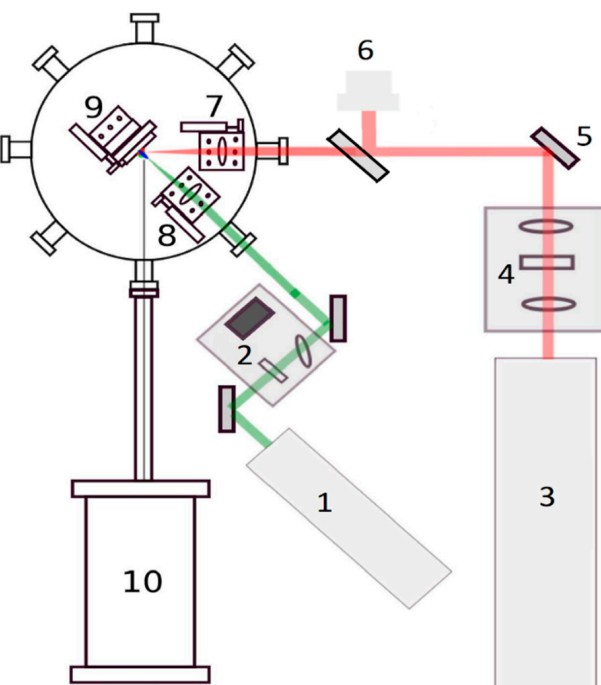

**Figure 1.** Experimental layout; 1: Nd:YAG laser, 2: Nd:YAG energy selector, 3: $CO_2$ laser, 4: plasma shutter, 5: steering mirror, 6: pulse profiler, 7: $CO_2$ focusing lens, 8: Nd:YAG focusing lens, 9: x, y, z target motion, 10: spectrometer.

The $CO_2$ pulse was shortened to remove the low energy tail using a plasma shutter based on the method pioneered by Hurst and Harilal; the shortened pulse had a FWHM of 21 ns and an energy of $\approx$100 mJ after the plasma shutter [11]. In this paper, only the shortened $CO_2$ pulse was used to form a plasma, it will hereafter be referred to as the $CO_2$ pulse.

The target was cleaned with the first laser shot to remove any contaminants left on the surface of the material; for all Nd:YAG spectra presented an average of five spectra recorded from the second shot was taken whereas for the $CO_2$, single shot LPP spectra are presented. The target was translated by 1 mm after each spectrum was captured. The chamber and spectrometer were pumped to a base pressure of $10^{-6}$ mbar.

The spectrometer is a grazing incidence JenOptik E-spec EUV spectrograph, which uses a Hitachi variable line spaced spherical grating (radius of curvature = 0.25 m) to form an image on a vertical plane (flat field effect). The spherical aberration corrected grating has 1200 lines per mm and forms an absolutely calibrated spectrum in the 9.8–18 nm region. The charge coupled device (CCD) camera of the spectrograph was cooled to $-13\,^{\circ}$C before spectra were captured.

The targets were prepared by melting weighed amounts of Pb and Sn in a dish using a hot plate to give target concentration ratios for Pb-Sn of 65–35 and 94–6 respectively. To check the target concentration before performing the experiment, the targets were sampled using a scanning electron microscopy (SEM) with energy dispersive X-ray spectroscopy (EDX), the results are presented in Table 1 and represent the average of 5 measurements and the variation in elemental composition. The errors in composition arise from the variation noted at different target locations and are reduced over the laser focal spot area. Given that average spectra from 5 shots were taken, the quoted mean composition represents a good estimation of the plasma elemental composition. In addition, the samples were also found to contain a small amount of oxygen and carbon on the surface. The composition for the alloy targets are relative Pb-Sn %, and do not include the % weight of oxygen and carbon that were present on the surface; whereas the % weight for the pure metals are with the inclusion of oxygen and carbon contaminants. The absence of any lines due to oxygen or carbon ions shows the effectiveness of the first laser shot in removing these species.

**Table 1.** Summary of the percentage weight of lead and tin in the samples and relative compositions of lead and tin in the alloy.

| Sn Percentage Weight [%] | Pb Percentage Weight [%] |
|:---:|:---:|
| 93 ± 4 | |
| | 84 ± 6 |
| 6 ± 1 | 94 ± 10 |
| 35 ± 26 | 65 ± 25 |

## 3. Theoretical Calculations

### 3.1. Collisional Radiative Model

Since both coronal equilibrium (CE) and local thermodynamic equilibrium (LTE) apply to the inner and outer regions of the plasmas respectively, it is best to adopt a model that incorporates both and also describes the region in between which corresponds to the bulk of a LPP. In this study the collisional radiative model (CR model) adapted by Colombant and Tonon (1973 [12] from the collisional radiative recombination coefficients of Bates *et al.* [13] is used). The CR model approximates to the CE model at low electron densities and the LTE model at high electron densities as the dominant recombination mechanism shifts from radiative recombination to three-body recombination with increasing electron density, the ion fractions are obtained on the assumption that the plasma is steady-state. This model provides total charge state densities for each ion, but it neglects the distribution of electrons amongst the excited states within an ion distribution. It also does not account for autoionization and dielectronic recombination but assumes that collisional ionization, radiative recombination, and three body

recombination dictate the population state of charge ($n_z$) at electron temperature ($T_e$). The plasma is therefore assumed to be optically thin to its own radiation; in this study the required ionization potentials were calculated using the Cowan code. According to the CR model, Equation (1) gives the electron temperature $T_e$ (eV) as a function of laser irradiance ($\phi$ in W cm$^{-2}$):

$$T_e(\text{eV}) \approx 5.2 \times 10^{-6} \, Z^{\frac{1}{5}} \left[ \lambda^2 \phi \right]^{\frac{3}{5}} \tag{1}$$

where $Z$ is the atomic number of the material and $\lambda$ is the wavelength of the laser in µm. Figure 2 shows the ion fractions for $T_e$ = 10 eV–100 eV for Pb (blue) and Sn (red) plasmas.

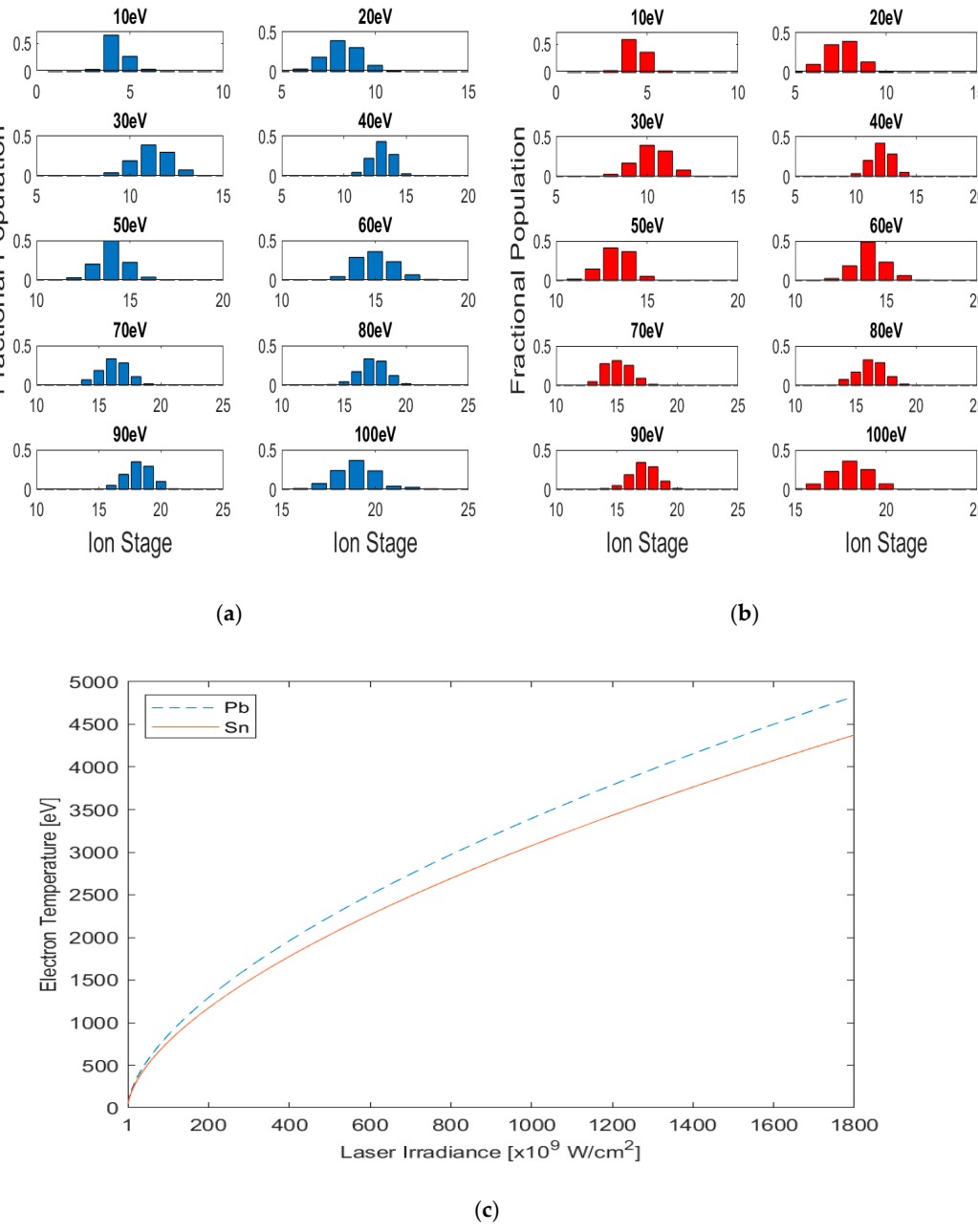

(a)
(b)

(c)

**Figure 2.** Ion stage populations for Pb (**a**) and Sn (**b**) plasmas for electron temperatures of 10 eV–100 eV. The bottom graph (**c**) shows electron temperature as a function of input laser irradiance.

### 3.2. Atomic Structure Calculations

Theoretical spectra for a large number of Sn and Pb ion stages were calculated with the CI Cowan RCN, RCN2 and RCG suite of codes [9]. For the calculations performed in this study; the average energy of the configurations was left unchanged as it was found to give good agreement with experiments to typically within 0.5 eV [14]. Since the calculated values of the Slater-Condon parameters are, in general, overestimated it is necessary to scale them, the scaling factor depends on the configuration of interest and approaches unity when the energy levels are close to hydrogenic. This can be understood as a requirement to take account of internal configuration perturbations not allowed for with a finite wavefunction basis or equivalently the effects of higher electron correlation on the level separation within a configuration [8]. The spectrum of Sn has been well documented in the literature [15–17] and the atomic transitions contributing to the emission in this region have been identified as being predominantly 4p–4d, 4d–4f and 4d–5p [4]. In the case of the Cowan code calculations of Sn, the spin orbit parameter was left unchanged while the $F^k$, $G^k$ and $R^k$ integrals were reduced by 25% of their ab initio values for Sn V and increased in steps to 85% of their ab initio values for Sn XX, the additional relativistic and correlation corrections were included for both the Sn and Pb calculations. The transitions included in the Sn calculation were the 4p–$m$d + $n$s ($m$ = 4–5, $n$ = 5–7) and 4d–$m$f + $n$p ($m$ = 4–8, $n$ = 5–7) for Sn V–Sn XIV; and 4s–$m$p ($m$ = 4–7) and 4p–$m$d + $n$s ($m$ = 4–7, $n$ = 5–7) for Sn XIV–Sn XX.

Figure 3 presents theoretical spectra calculated with the Cowan code for Sn V–Sn XX ions, the weighted transition probabilities (gA values) were convolved with a Gaussian of width 0.01 nm and normalized in the 9.8–18 nm region. The 4d–4f transitions first appear as the feature on the far right in Sn VII, and following the onset of excited state CI of the type $4p^6 4d^N 4f + 4p^5 4d^{N+1}$ at Sn VI shift to shorter wavelength with increasing ionization. For Sn XI the 4d–5p transitions can be seen at 16 nm, which also shift to shorter wavelength with increasing ionization to overlap with the 4p–4d and 4d–4f lines in Sn XIII. From Sn XV onwards the emission observed is solely from 4p–4d excitation in ions with a $4p^N$ ground configuration, many lines of which have been recently identified [18,19].

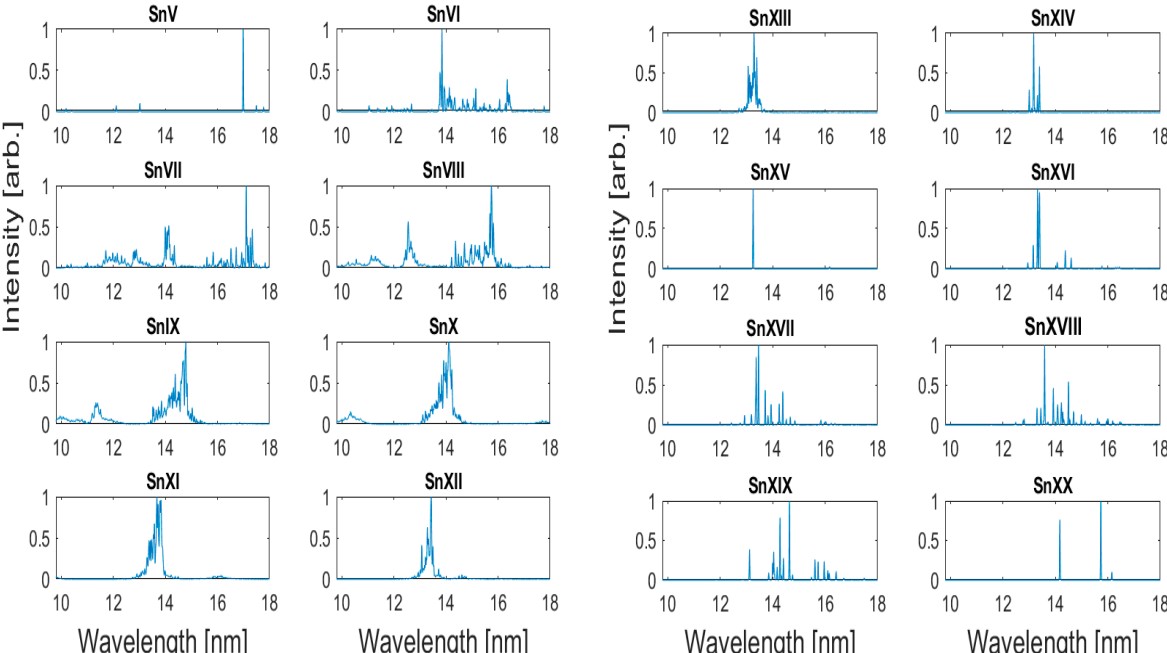

**Figure 3.** Theoretical spectra calculated with the Cowan suite of codes for Sn V–Sn XX ions, weighted transition probabilities are convolved with a Gaussian of width 0.01 nm and normalized in the 9.8–18 nm region. (See text.).

The emission from Pb in this region is less well documented; earlier work by Bridges *et al.* identified broad UTA features at around 12.5 nm and 16 nm with a dip in the spectrum near 14 nm [20].

Churilov *et al.* identified 9 lines from $5s^25p^65d – (5s^25p^55d6s + 5s^25p^67p)$ transitions of PbXIV [21]. In a later study Churilov *et al.* identified 57 lines from $5p^65d – (5p^65f + 5p^66p + 5p^55d^2 + 5p^55d6s)$ transitions in Pb XIV between 9.9 and 20.5 nm [22]; while Kaufman *et al.* identified 6 lines from $5p^64f^{14}$ – $(5p^64f^{13}5d^1 + 5p^55d^1 + 5p^56s)$ transitions in Pb XV [23].

In other related work by Carroll *et al.* [24,25] LPP spectra between 7 nm and 13.5 nm from uranium and thorium were found to be dominated by 5d–5f transition array emission which peaks around 9.5 nm and 10.5 nm respectively with a second UTA observed in the thorium spectr a at 8–9.5 nm primarily due to 5p–5d transitions. Liu *et al.* has identified emission bands in Bi plasmas between 8 nm and 17 nm from 5p–5d, 5d–5f, 5s–5p and 5f–5g transitions in ion stages from $Bi^{23+}$–$Bi^{35+}$ [26]. In other related work Liu *et al.* carried out a study to assess the effects of CI on the UTA emission from $5p^65d^{N+1} – 5p^55d^{N+2} + 5d^N5f^1$ transitions in elements Au to U [27].

The line emission from Pb in the 9.8–18 nm region for the range of laser power densities applied originates predominantly from ions with open 5d and 5p subshells. In the case of Pb VII–Pb XIV, emission predominantly originates from 5d–5f, 5p–5d, 5d–6p and 5p–6s transitions. For Pb XV–Pb XX the emission originates from 5p–5d and 5s–5p transitions. Figure 4 shows the calculated positions of 5p–6d (cyan), 5d–6f (magenta), 4f–5d (black), 5p–5d (red), 5d–5f (blue), 5d–6p (yellow), 5p–6s (green) for Pb VI–Pb XIV and 5s–5p (magenta), 5p–5d (red) for Pb XV–Pb XX, plotted as a function of increasing ionization. With increased ionization the emission can be seen to move slightly to higher energy. Each transition array is normalized against the highest gA value for that array in the 9.8–18 nm region.

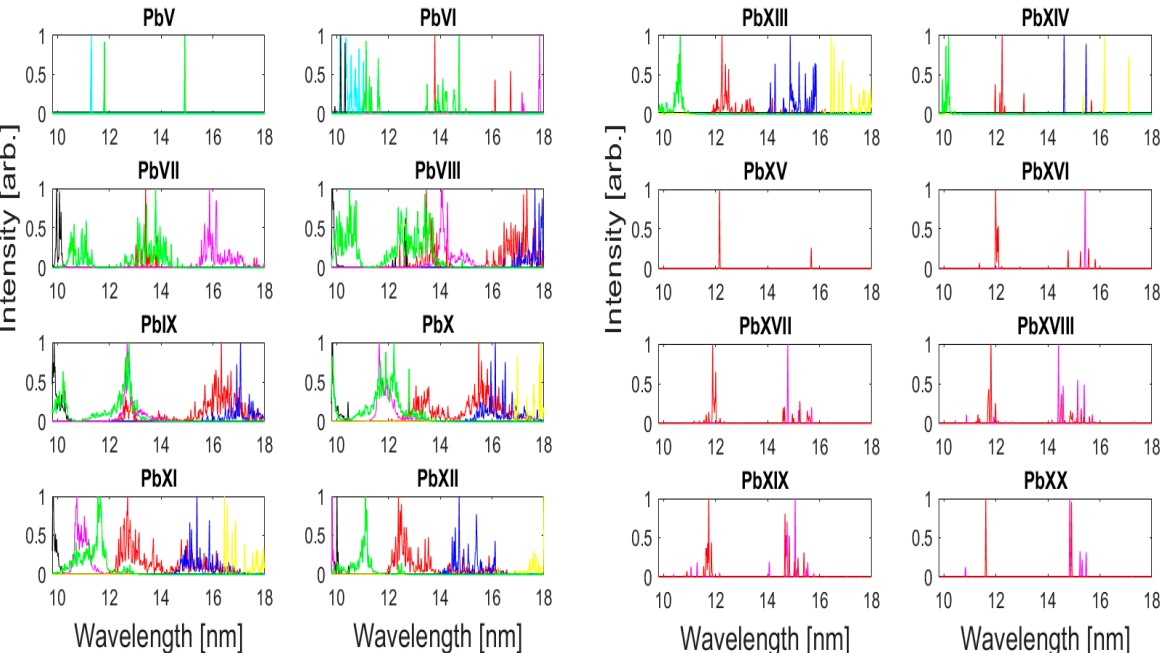

**Figure 4.** Relative intensity versus wavelength for 5p–6d (cyan), 5d–6f (magenta), 4f–5d (black), 5p–5d (red), 5d–5f (blue), 5d–6p (yellow), 5p–6s (green) for Pb VI–Pb XIV and 5s–5p (magenta), 5p–5d (red) for Pb XV–Pb XX. The weighted transition probabilities are convolved with a Gaussian of width 0.01 nm and while each transition array is normalized against the highest gA value for that array in the 9.8–18 nm region. (Colour online; See text.)

The transitions included in the calculation were the 4f–5d, 5d–*m*f (*m* = 5–7) + *n*p (*n* = 6–7) and 5p–*m*d (*m* = 5–7) + *n*s (*n* = 6–7) for Pb V–Pb XIV, (where the $4f^{13}5s^25p^65d^N6s^1$ complex configuration was included to allow for the effects of mixing with the $5p^55d^N6s^1$ configuration); 5p–*m*d + *n*s (*m*=5–7, *n*= 6–7), 4f–*m*d (*m*= 5–6) and 5s–*m*p (*m*= 5–6) for Pb XV–PbXX. The configurations included in the calculations have been limited to the most important ones, while doubly excited configurations have been neglected to limit calculation time; it is well known that Δn ≥ 2 transitions are, in general,

collisionally quenched in LPPs with an electron density of $10^{21}$ cm$^{-3}$ and are therefore not included in the theoretical spectra presented. Figure 5 presents theoretical spectra calculated with the Cowan code for Pb V–Pb XX ions, the weighted transition probabilities (gA values) were convolved with a Gaussian of width 0.01 nm and normalized in the 9.8–18 nm region.

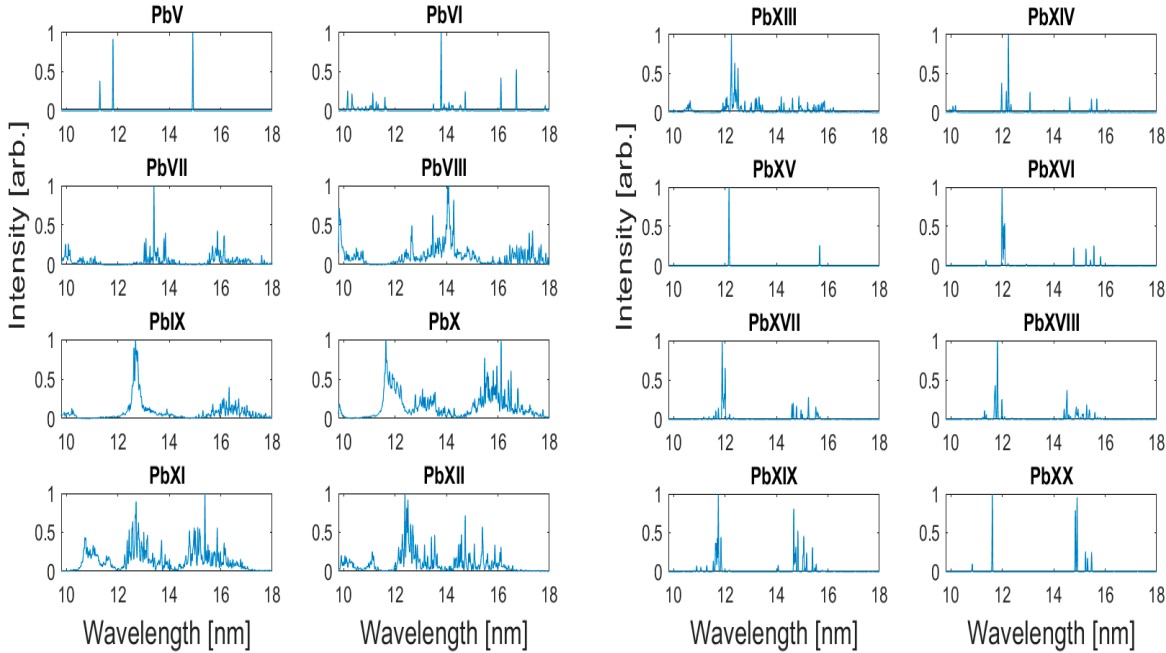

**Figure 5.** Theoretical spectra calculated with the Cowan code for Pb V–Pb XX ions, the weighted Table 0. nm and normalized in the 9.8–18 nm region. (See text.).

In these calculations the spin orbit parameters were again left unchanged while the $F^k$, $G^k$ and $R^k$ integrals were reduced by 13% for Pb V to give the best agreement with the $5d^96p^1$ energy levels identified by Goldsmith and Conway [28]. Generally, a reduction in the scaling of $F^k$, $G^k$ and $R^k$ with increasing ionization is necessary to obtain agreement between calculated and experimental spectra. However, it was found that leaving the values of $\zeta$ unchanged, and reducing $F^k$, $G^k$ and $R^k$ by 13% gave the best agreement with the $5s^25p^65d$ – ($5s^25p^55d6s$ + $5s^25p^67p$) lines of Pb XIV observed by Churilov and Joshi [21]. For Pb XV, $\zeta$ value was left unchanged while the $F^k$, $G^k$ and $R^k$ parameters were reduced by 20% to match the $5p^64f^{14}$ – ($5p^64f^{13}5d^1$ + $5p^55d^1$ + $5p^56s$) lines identified by Kaufman and Sugar [23], no extrapolation beyond Pb XV was carried out and the scaling of the Slater integrals was left unchanged for higher ion stages. Regarding the lines identified by Kaufman and Sugar it should be noted that the $5p^6$ – $5p^55d$ $(3/2,5/2)_1$ line observed at 14.4252 nm in Pb XV was calculated to lie at 15.679 nm, while for all the various scaling ratios tried better agreement was observed if their identification for the $4f^{14}$–$4f^{13}5d$ $(5/2,5/2)_1$ at 7.9564 nm and $5p^6$–$5p^56s$ $(1/2,1/2)_1$ at 7.9959 nm were interchanged; unfortunately, neither of these lines are observed experimentally in the current study.

With increasing ionization, the coupling scheme tends to jj but most commonly, the electronic structure will be best represented by a coupling scheme intermediate between the two extremes of LS or jj, as in general neither the spin-orbit or Coulomb interaction will ever go to zero. In the current study it was found that a higher eigenvector purity could be obtained using a Jj coupling nomenclature, as opposed to the LS coupling labels used by Churilov and Joshi. To remove the ambiguity involved in Jj coupling labels the ground states are labelled with an LS term, as regards the excited states the assignment of the core electrons is done using an LS term for the leading eigenvector followed by a J value for the core electrons, with a j value for the outer subshell being presented and the usual J subscript for the eigenstate.

## 4. Results

### 4.1. Effects of Input Laser Energy on Continuum Emission in the 10–18 nm Region

Figures 6–10 show spectra recorded with the Nd:YAG laser for laser focal spots radii of 41 µm (1 mm from focus), 100 µm (2.5 mm from focus), 200 µm (5 mm from focus), 300 µm (7.5 mm from focus) and 400 µm (10 mm from focus); and energy values of 185 mJ, 270 mJ, 350 mJ and 450 mJ at the target; the power densities range from $7.3 \times 10^9$ W cm$^{-2}$–$1.2 \times 10^{12}$ W cm$^{-2}$. The focal spot sizes where estimated assuming a Gaussian laser beam and employing the Rayleigh length formula. The dip in the experimental spectrum from 12.36–12.43 nm is due to an error in the intensity calibration of the spectrograph at this wavelength which corresponds to the silicon L-edge. It is present in all spectra and hereafter will be ignored.

As can be seen from Figure 6 to Figure 9, an increase in laser input energy corresponds to an increase in continuum emission. This can be explained by a greater ablated mass, which scales according to $\phi^{5/9}$ for a fixed pulse duration resulting in a larger number of emitting ions present in the plasma [29,30]. Cummings *et al.* performed time dependent spatially resolved calculations of ion and electron density, and found that increasing the laser energy creates a plasma with a shorter high density gradient and a longer low density gradient that reduces self-absorption [31]. The overall peak of the emission also shifts to shorter wavelength for Pb as the 5d subshell empties and the 5p subshell emission takes over. Absorption within the emission band region also increases due to an increase in free electrons available in the outer plasma to repopulate ions of a lower charge state; this is most evident in the Sn spectra.

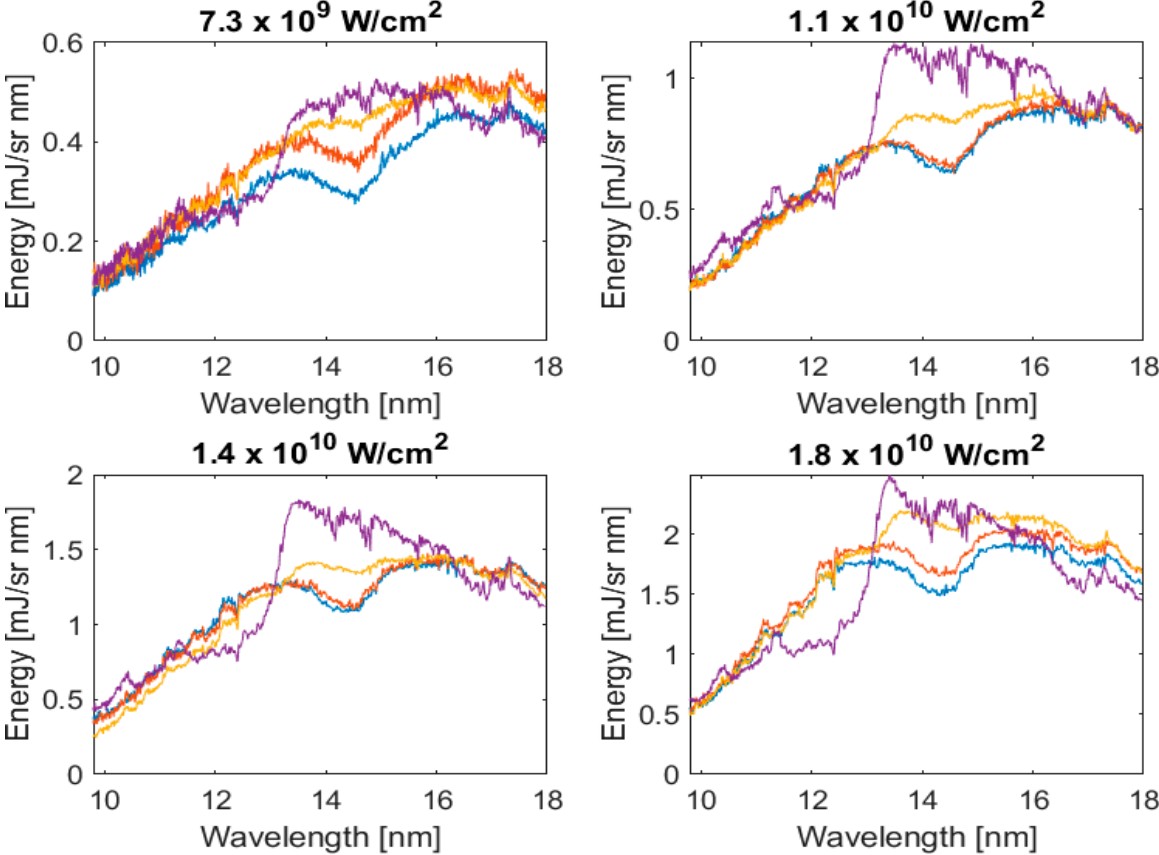

**Figure 6.** Spectra recorded for a fixed spot radius of 400 µm; the laser energies were 185 mJ, 270 mJ, 350 mJ, 450 mJ; blue: Pb, red: PbSn946, yellow: PbSn6535 and purple: Sn.

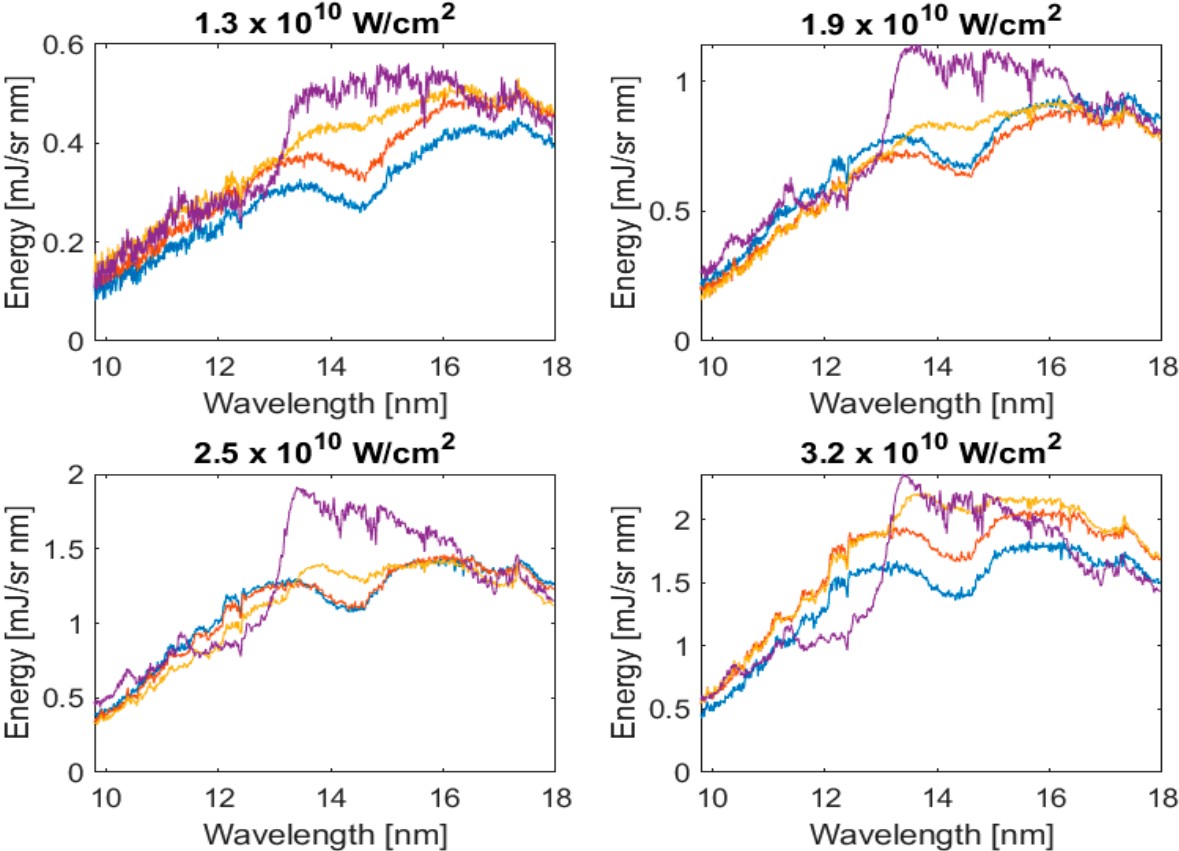

**Figure 7.** Spectra recorded for a fixed spot radius of 300 μm; the laser energies were 185 mJ, 270 mJ, 350 mJ, 450 mJ; blue: Pb, red: PbSn946, yellow: PbSn6535 and purple: Sn.

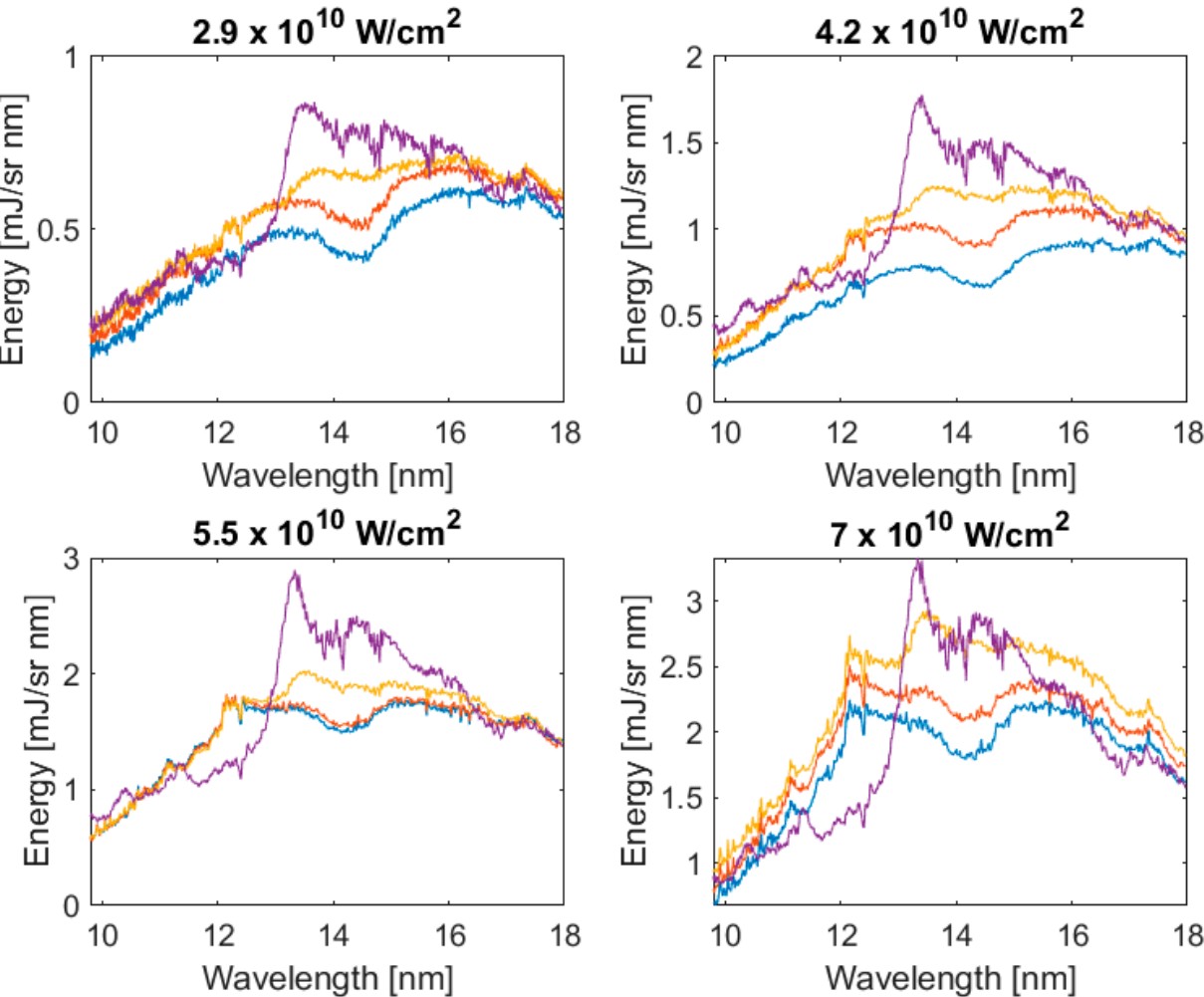

**Figure 8.** Spectra recorded for a fixed spot radius of 200 μm, the laser energies were 185 mJ, 270 mJ, 350 mJ, 450 mJ; blue: Pb, red: PbSn946, yellow: PbSn6535 and purple: Sn.

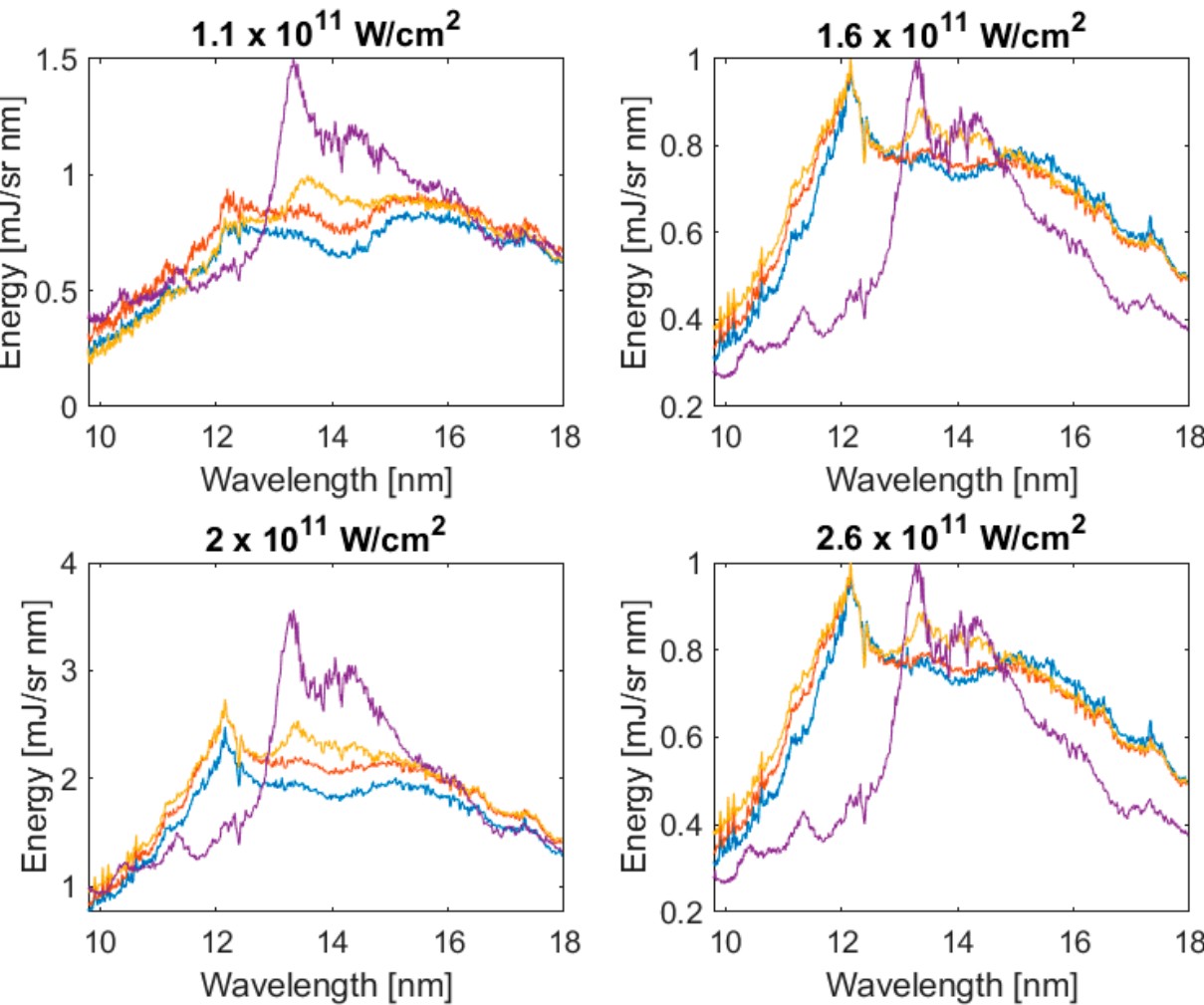

**Figure 9.** Spectra recorded for a fixed spot radius of 100 μm, the laser energies were 185 mJ, 270 mJ, 350 mJ, 450 mJ; blue: Pb, red: PbSn946, yellow: PbSn6535 and purple: Sn.

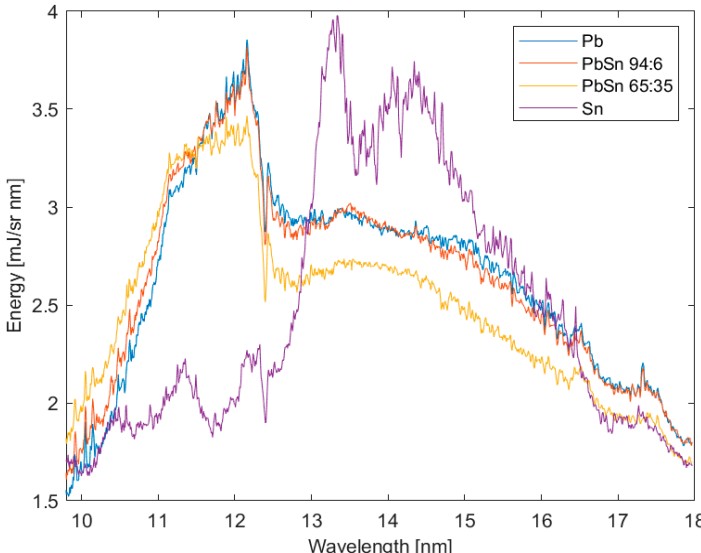

**Figure 10.** Spectra recorded for a fixed spot radius of 41 μm and laser energy of 450 mJ, $1.2 \times 10^{12}$ W cm$^{-2}$; blue: Pb, red: PbSn946, yellow: PbSn6535 and purple: Sn.

One of the most noticeable differences between the Pb and Sn spectra is the growth in spectral efficiency with increasing laser energy for a fixed spot size and the lack of discrete structure due to emission and/or absorption in the Pb plasma. For Sn, absorption features appear at all power densities with the longer wavelength absorption from Sn VII–Sn IX decreasing, while absorption from Sn X and Sn XI becomes more pronounced with increasing power density and average charge in the plasma plume ($\xi_{Av}$). For Pb the absorption features are much less obvious and are hidden in the continuum; the most dominant absorption features appear between 15–16.5 nm and are present in all spectra. Compared to medium Z elements, high Z elements have been shown to have a more intense continuous spectrum in the EUV region due to the increase in free-free and free-bound radiation, with broad transition array (TA) emission structures superimposed on the continuum resembling bumps as seen here in Pb [32].

Of particular interest is the absence of a pronounced feature due to Sn UTA emission near 13.5 nm in the alloy spectra recorded with the Nd:YAG laser. It is well known that in spectra of Sn plasmas created on targets where the elements other than Sn are low Z elements such as Sn salts or $SnO_2$ dissolved in plastics that the peak of the Sn UTA emission increased in intensity with decreasing concentration and reached a maximum at a concentration of ~2% due to the decrease in opacity of the highest gA value transitions. For pure Sn targets the plasma is optically thick as evidenced here by the strong absorption features which can be clearly attributed to 4–4 transitions in Sn VII–Sn XI depending on the plasma temperature. Based on these observations it could be expected that the Sn UTA intensity should increase in the alloy, in particular, in that containing a 6% concentration. However, in all of the spectra shown, the emission from the alloys at 13.5 nm is consistently lower than that from pure Sn. Indeed, the emission from the alloys appears to be very similar to that of pure Pb, especially in the case of a 6% concentration and only for the 35% Sn concentration does a weak peak appear in the Sn UTA region. None of the alloy spectra contain features due to either discrete emission or absorption associated with strong individual 4d–4f or 4p–4d lines in Sn and this is particularly evident from Figure 10. Thus the expected increase in UTA emission appears to be quenched by strong 5p–5d absorption in the Pb ions.

It is particularly instructive to compare the absolute emission of the alloys with that of a spectrum generated by combining the emission of pure Sn and Pb summed according to their ratios in the alloys for both Nd:YAG and $CO_2$ laser produced spectral emission, as shown in Figures 11 and 12. The Nd:YAG spectra of the pure metals are optically thick and one would expect that both the overall intensity should increase and discrete structure such as UTA emission should become more pronounced in the alloy emission due to the reduction in concentration. The overall intensity does indeed increase but the spectra actually appear flatter with less evidence of discrete structure. Indeed, for the spectra presented in Figure 11 the overall emission is most intense for the 65–35 combination where the Pb concentration is lowest. For the $CO_2$ LPPs, which are essentially optically thin, while there is an increase again in overall emission from the alloys, the UTA at 13.5 nm is particularly clear in the 35% spectrum and essentially mirrors the Sn concentration.

Pure continuum emission originates from recombination and bremsstrahlung, the latter scales with $\xi_{Av}^2$, while the former scales as $\xi_{Av}^4$ [33]. The $\xi_{Av}$ for Pb is higher than for Sn as ionization potentials decrease for a given charge state with increasing Z along the periodic table and a heavier ion mass results in increased time spent in the deflagration zone [34]. The peak of bremsstrahlung emission corresponds to $\lambda_B = 620/T_e$ nm [35]; at an irradiance of $5.5 \times 10^{10}$ W cm$^{-2}$ for a Pb LPP, $T_e = 38$ eV, $\xi_{Av} = 13+$, $\lambda_B = 16.3$ nm, for a Sn LPP, $T_e = 34$ eV, $\xi_{Av} = 11+$, $\lambda_B = 18.2$ nm corresponding to the longer wavelength limit on the spectra captured.

However, the spectral efficiency is low for Pb compared to Sn at a laser energy of 185 mJ regardless of spot size. If the difference in emission between the Pb and Sn was simply due to continuum emission for a higher $\xi_{Av}$ in the LPP, the spectrum at an irradiance of $1.1 \times 10^{11}$ W cm$^{-2}$ (spot radius 100 μm) for Pb would be expected to have a greater intensity. For Pb, $T_e = 57$ eV, $\xi_{Av} = 15+$ ($\xi_{Av}^4 = 50,625$),

$\lambda_B = 10.9$ nm; for Sn, $T_e = 52$ eV, $\xi_{Av} = 14+$ ($\xi_{Av}^4 = 38,416$), $\lambda_B = 12$ nm, approximately 25% more continuum emission is expected for Pb.

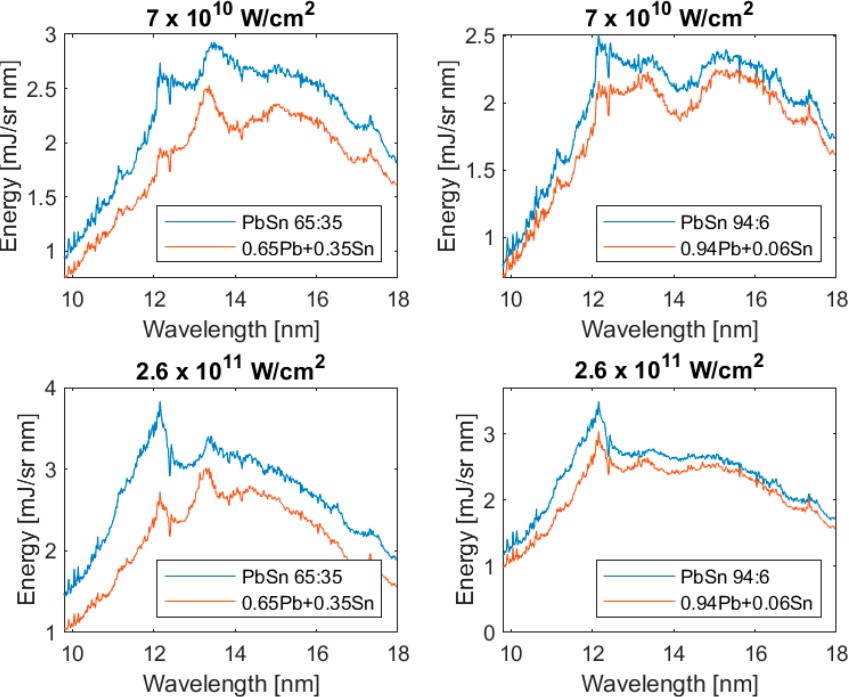

**Figure 11.** Spectra of Nd:YAG produced plasmas for the alloys compared with those of pure Sn and Pb spectra multiplied by their respective ratios in the alloy targets and summed for laser irradiance of $7 \times 10^{10}$ W cm$^{-2}$ and $2.6 \times 10^{11}$ W cm$^{-2}$.

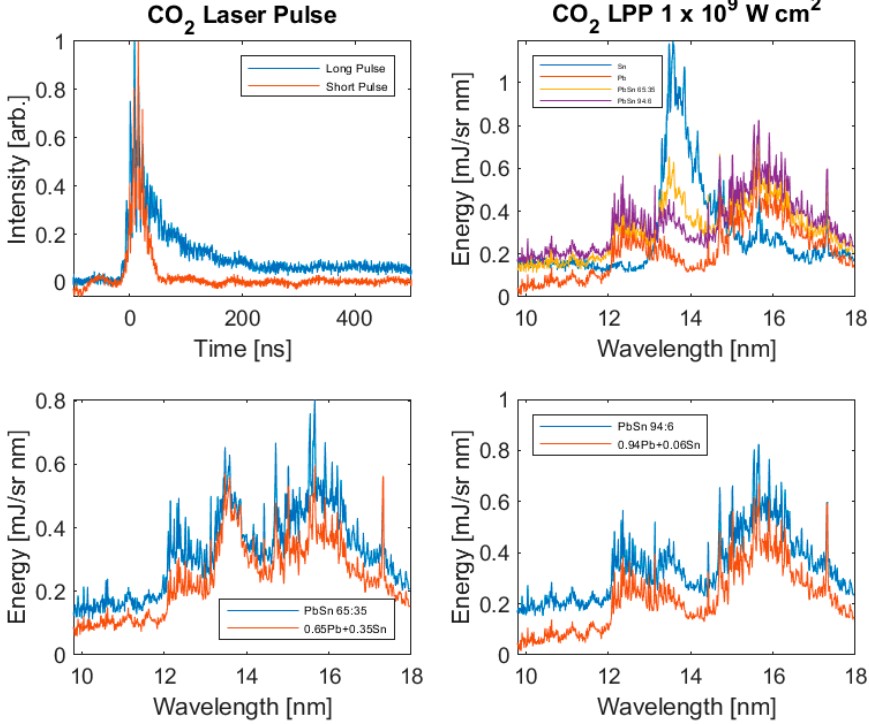

**Figure 12.** The shortened $CO_2$ pulse in seen in the top left, the captured spectra are presented in the top right. In the bottom left and right the pure Sn and Pb spectra are multiplied by their respective ratios in the alloy targets and summed. For Pb $T_e = 54$ eV and $\xi_{Av} = 15+$, while for Sn $T_e = 49$ eV and $\xi_{Av} = 13+$.

### 4.2. Line Emission in the 10−18 nm Region

The contribution from line emission in the region for Sn (red) and Pb (blue) plasmas for $T_e$ = 10 eV to 100 eV are shown in Figure 13. To obtain these synthetic spectra the theoretical gA values were again weighted assuming a Boltzmann population distribution amongst the levels of each excited configuration and a level degeneracy of 2J+1. The ion fractions for each $T_e$ are calculated using the CR model and the resultant line strengths for each transition are convolved with a Gaussian of width 0.01 nm and summed to give the total emission from a plasma at a single temperature. As seen from Figure 13 4d–4f transitions in Sn (red) create a UTA with emission centered around 13.5 nm. The shift towards shorter wavelength of the 4d–4f array is the result of 4f wavefunction contraction with increasing ion stage [34–36].

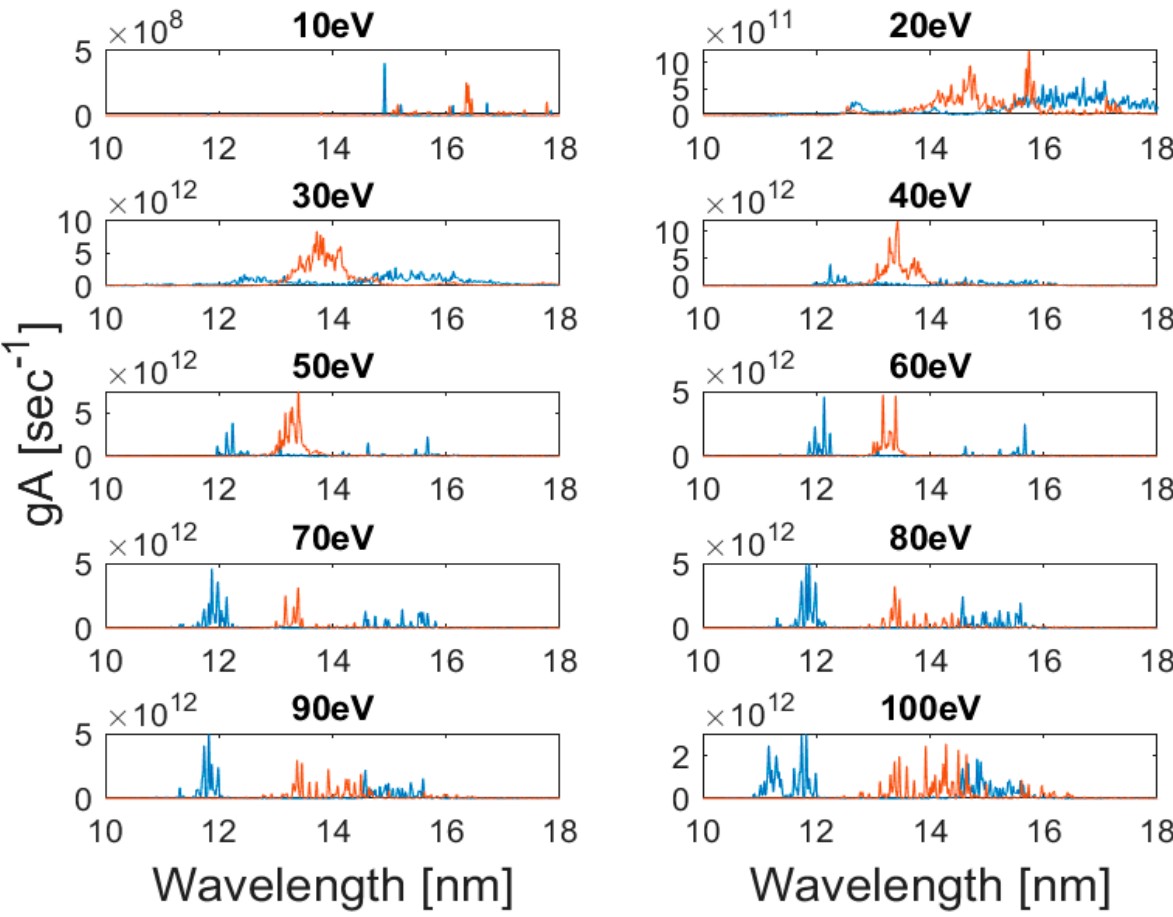

**Figure 13.** The contribution from line emission in the region for Sn (red) and Pb (blue) plasmas from 10 eV to 100 eV.

Liu *et al.* previously showed that the effect of CI on the 5d–5f and 5p–5d emission arrays along the isoelectronic sequence becomes important for 5d–5f beyond $Pb^{7+}$, while the effects of CI on 5p–5d emission only becomes important once these transitions overlap with the 5d–5f array [27]. Overall, they found that CI causes a shift to higher energy and a slight increase in array width. Changing the principal quantum number from 5 to 4 gives the configurations leading to the Sn UTA, where emission arrays have a greater overlap and where CI is known to cause a shift to higher energy and a large decrease in array width [37]. Figure 14 shows the positions of calculated Pb energy levels, the importance of including various configurations becomes more obvious from the overlap of their configuration energies. Note the experimental emission range of 69 eV to 127 eV (9.8 nm–18 nm) is marked with red lines in Figure 14. The energy levels are so close that they appear to form a continuous band, the number of levels increases to a maximum and decreases in proportion to the statistical weight

(g) of the subshell; g = [(4l + 2)!/w!(4l + 2 − w)!], where l is the orbital angular momentum number and w is the occupancy of the orbital [9]. The overlapping energy levels of the $5p^55d^{N+1}$, $5d^{N-1}5f^1$ and $5d^{N-1}6p^1$ configurations at lower energies and the $5p^56s^1$, $5d^{N-1}6f^1$ and $5d^{N-1}7p^1$ at higher energies is significant, warranting their inclusion. The $5p^55d^{N+1}$, $5d^{N-1}5f^1$ and $5d^{N-1}6p^1$ configurations do not coalesce to the same degree as the corresponding 4d configurations in Sn ions, with the corresponding emission bands showing greater separation over the range of ion stages considered here. This is especially evident in the $5p^55d^{N+1}$ emission to the ground configuration, where the emission array splits into two sub-arrays due to spin-orbital interaction [38]. The preceding section explains the difference in the Pb and Sn calculated spectra, where Pb is seen to have a much broader emission structure compared to Sn.

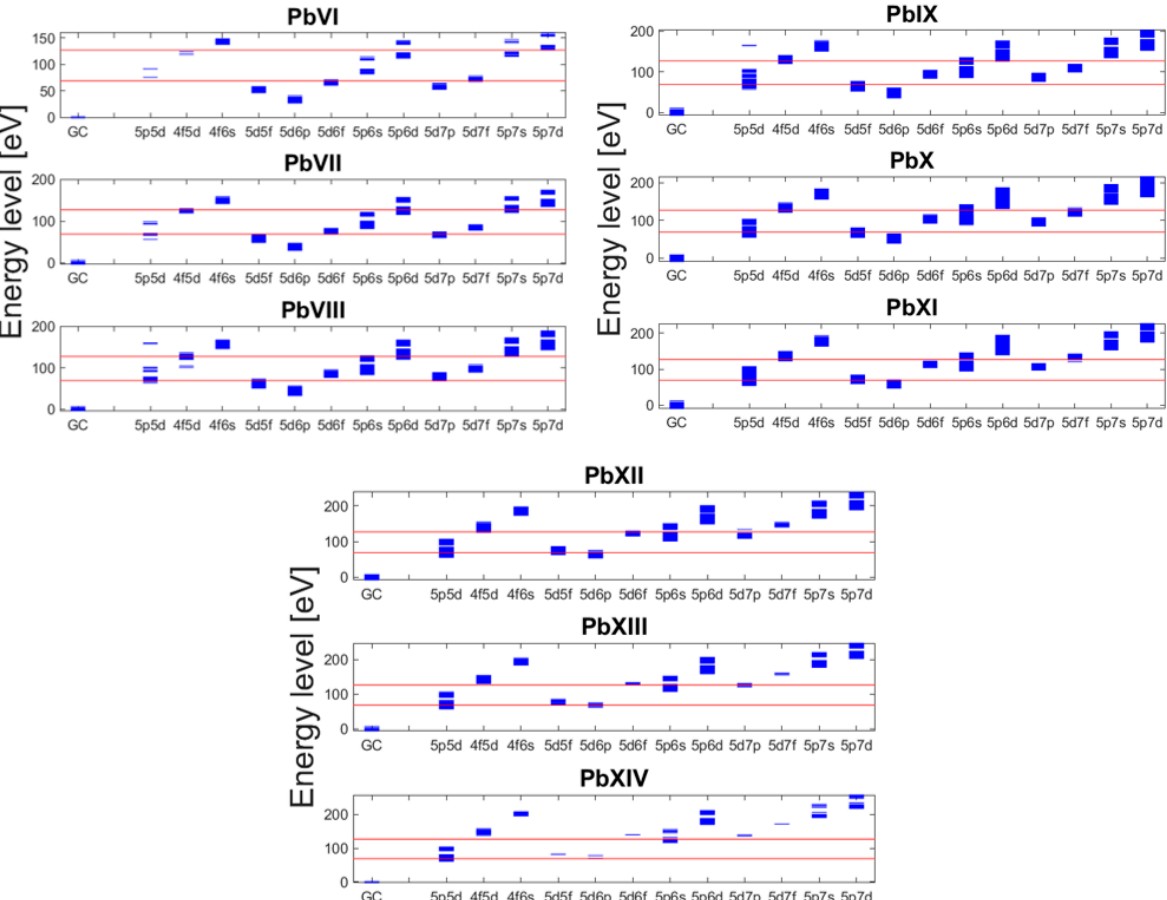

**Figure 14.** Relevant energy levels of excited configurations in Pb VI to Pb XIV taken from the Cowan code calculations; GC represents the ground configuration, $5d^9$ to $5d^1$. The red lines indicate the approximate region of interest. (See text.).

As stated earlier in calculating the line emission, the plasma was assumed to be optically thin, which a Nd:YAG plasma clearly is not. Previous work by Yan-Biao *et al.* [39] has shown that dielectronic recombination can significantly alter the ion stage population for an optically thick plasma; while work by Sasaki *et al.* [40,41] has shown the contribution from satellite lines increases with opacity of the resonance lines, leading to a broadening of a UTA and the quasicontinuum nature of the observed spectra.

Looking at the difference in the Pb and Sn plasma dynamics, Ahmed *et al.* found that molybdenum LPPs have a slight increase in lateral to normal expansion compared to aluminium LPPs, Mo has slower moving ions and an emission flux that peaks at 40° to the target surface [42]. This would indicate

that the heavier Pb ions form a denser plasma, thus increasing self-absorption and that greater lateral expansion of the plasma reduces coupling with the laser.

### 4.3. Discussion of Spectra and Line Identification

Plasmas produced at the lower power densities of $7.3 \times 10^9$ W cm$^{-2}$ and $1.1 \times 10^9$ W cm$^{-2}$ appear to have ion stages up to a maximum of Pb X present. At $1.4 \times 10^{10}$ W cm$^{-2}$ a number of small peaks between 13 and 15.5 nm corresponding to $5p^65d^5 - 5p^55d^6$ emission from Pb X and $5p^65d^4 - 5p^55d^5$ emission from Pb XI begin to appear; beyond this, line absorption around 16 nm is present with preliminary analysis suggesting $5p^65d^5 - 5d^45f^1$ transitions in Pb X. At shorter wavelengths the emission feature at 10.4 nm in the alloy targets corresponds to $4p^64d^5 - 4d^45f^1 + 4d^46p^1$ emission from Sn X ions [15]. For the Pb plasmas the emission structure on the shorter wavelength side of 12.4 nm become more defined and corresponds to $5p^6 - 5p^56s^1$ transitions in Pb XI and Pb XII. Three peaks are observed at 11.14 nm, 11.16 nm and 11.2 nm and give good agreement with $5p^65d^3 - 5p^55d^36s^1$ transitions of Pb XII; $^2G_{9/2} - (6,1/2)_{11/2}$, $^2H_{11/2} - (7,1/2)_{13/2}$ and $^4F_{3/2} - (1,1/2)_{3/2}$ respectively. All lines observed are summarized in Table 2.

For a power density of $2.5 \times 10^{10}$ W cm$^{-2}$ the $5p^65d^1 - 5p^55d^16s^1$ lines of Pb XIV; 9.91 nm $^2D_{5/2} - (3,1/2)_{5/2}$, 9.99 nm $^2D_{5/2} - (3,1/2)_{7/2}$, 10.02 nm $^2D_{5/2} - (2,1/2)_{5/2}$, 10.05 nm $^2D_{3/2} - (2,1/2)_{5/2}$, 10.12 nm $^2D_{5/2} - (2,1/2)_{3/2}$, 10.15 nm $^2D_{3/2} - (3,1/2)_{5/2}$ and 10.2 nm $^2D_{5/2} - (4,1/2)_{7/2}$ are visible but weak, becoming apparent at the power density of $1.6 \times 10^{11}$ W cm$^{-2}$; these peaks can be seen in Figure 15 where the highest power density for each spot is graphed between 9.8 and 12.5 nm.

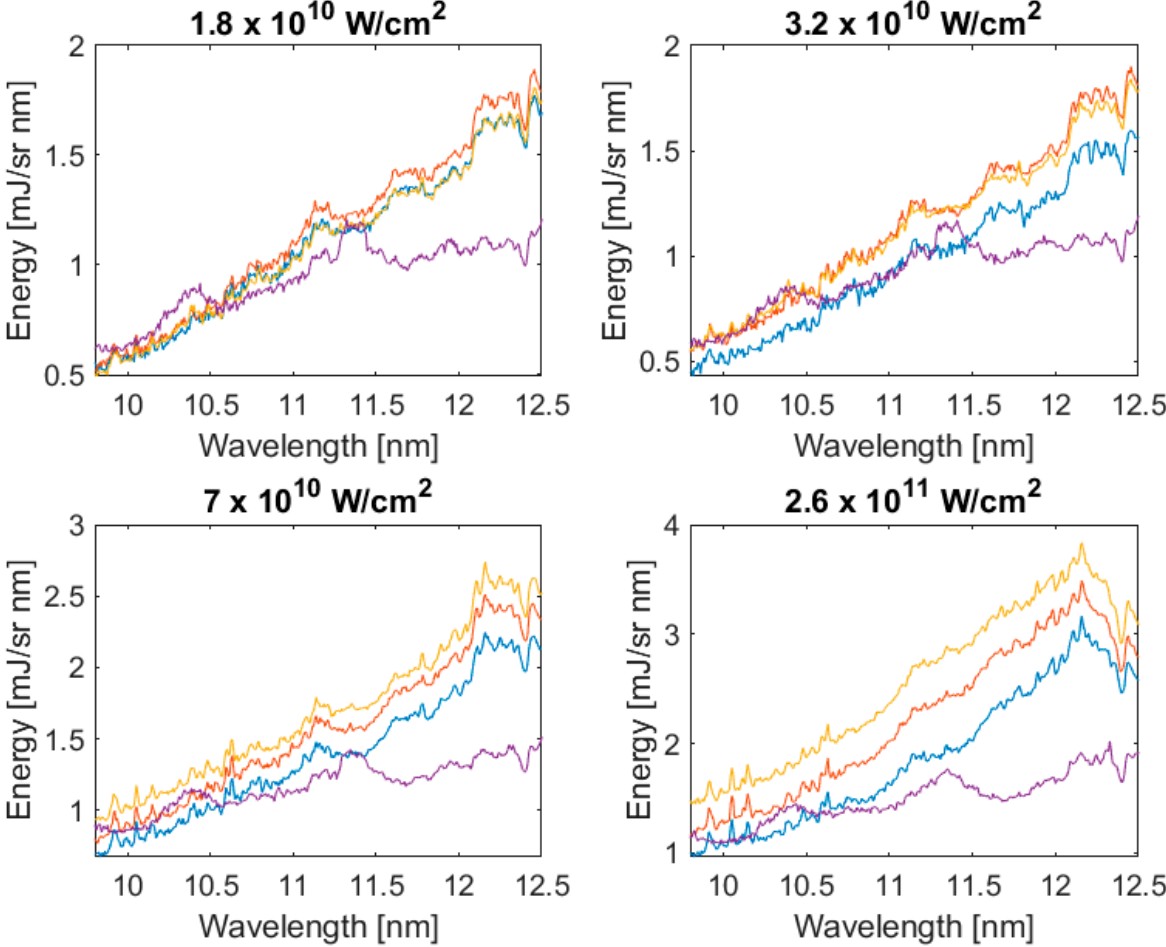

**Figure 15.** Spectra recorded at the highest power density for spot radii of 100 μm, 200 μm, 300 μm and 400 μm between 9.8 and 12.5 nm; blue: Pb, red: PbSn946, yellow: PbSn6535 and purple: Sn.

**Table 2.** Observed transitions along with calculated wavelengths and weighted transition probabilities determined from the Cowan suite of codes. The previously observed lines are from the work by Churilov *et al.* (C) [21,22] and Kaufman *et al.* (K) [23].

| Ion Stage | Transition Configuration | Jj Composition | Observed λ/nm | Calculated λ/nm | (×10$^{11}$) gA/s$^{-1}$ | Prev. Observed λ/nm |
|---|---|---|---|---|---|---|
| Pb XIV | $5p^6 5d^1$ ($^2D_{5/2}$) − $5p^5 5d^1 6s^1$ | 99% ($[^2D_{5/2}]$ 3,1/2)$_{5/2}$ | 9.91 | 9.950 | 4.98 | 9.926 C |
| Pb XIV | $5p^6 5d^1$ ($^2D_{5/2}$) − $5p^5 5d^1 6s^1$ | 98.7% ($[^2D_{5/2}]$ 3,1/2)$_{7/2}$ | 9.99 | 10.023 | 1.78 | 10.003 C |
| Pb XIV | $5p^6 5d^1$ ($^2D_{5/2}$) − $5p^5 5d^1 6s^1$ | 95.2% ($[^2D_{3/2}]$ 2,1/2)$_{5/2}$ | 10.02 | 10.079 | 4.15 | 10.063 C |
| Pb XIV | $5p^6 5d^1$ ($^2D_{3/2}$) − $5p^5 5d^1 6s^1$ | 48.6% ($[^2D_2]$ 2,1/2)$_{5/2}$ + 45% ($[^2D_3]$ 3,1/2)$_{5/2}$ | 10.05 | 10.087 | 6.38 | 10.070 C |
| Pb XIV | $5p^6 5d^1$ ($^2D_{5/2}$) − $5p^5 5d^1 6s^1$ | 91.2% ($[^2D_2]$ 2,1/2)$_{3/2}$ | 10.12 | 10.143 | 3.07 | 10.132 C |
| Pb XIV | $5p^6 5d^1$ ($^2D_{3/2}$) − $5p^5 5d^1 6s^1$ | 54.1% ($[^2D_3]$ 3,1/2)$_{5/2}$ −42.8% ($[^2D_2]$ 2,1/2)$_{5/2}$ | 10.15 | 10.151 | 1.56 | 10.137 C |
| Pb XIV | $5p^6 5d^1$ ($^2D_{5/2}$) − $5p^5 5d^1 6s^1$ | 97.4% ($[^2D_4]$ 4,1/2)$_{7/2}$ | 10.2 | 10.179 | 9.22 | 10.176 C |
| Pb XIII | $5p^6 5d^2$ ($^1G_4$) − $5p^5 5d^2 6s^1$ | 62.7% ($[^1G_4]$ 9/2,1/2)$_4$ −35.8% ($[^3F_4]$ 9/2,1/2)$_4$ | 10.48 | 10.495 | 6.07 | |
| Pb XIII | $5p^6 5d^2$ ($^3F_3$) − $5p^5 5d^2 6s^1$ | 88.4% ($[^3F_3]$ 9/2,1/2)$_4$ −3% ($[^3F_2]$ 7/2,1/2)$_4$ | 10.6 | 10.607 | 9.81 | |
| Pb XIII | $5p^6 5d^2$ ($^3F_4$) − $5p^5 5d^2 6s^1$ | 50.8% ($[^3F_4]$ 11/2,1/2)$_5$ − 44.3% ($[^1G_4]$11/2,1/2)$_5$ | 10.63 | 10.648 | 11.90 | |
| Pb XII | $5p^6 5d^3$ ($^2G_{9/2}$) − $5p^5 5d^3 6s$ | 38.8% ($[^2G_{9/2}]$ 6,1/2)$_{11/2}$ −29.6% ($[^2H_{9/2}]$6,1/2)$_{11/2}$ −22.6% ($[^4F_{9/2}]$ 6,1/2)$_{11/2}$ | 11.14 | 11.133 | 12.21 | |
| Pb XII | $5p^6 5d^3$ ($^2H_{11/2}$) − $5p^5 6s$ | 90.2% ($[^2H_{11/2}]$7,1/2)$_{13/2}$ | 11.16 | 11.162 | 10.45 | |
| Pb XII | $5p^6 5d^3$ ($^4F_{3/2}$) − $5p^5 5d^3 6s$ | 31.5% ($[^4F_{1/2}]$ 1,1/2)$_{3/2}$ + 25.7%($[^4F_{1/2}]$2,1/2)$_{3/2}$ + 28.1% ($[^2D_{1/2}]$2,1/2)$_{3/2}$ | 11.20 | 11.195 | 3.80 | |
| Pb XV | $5p^6$ ($^1S_0$)−$5p^5 5d^1$ | 90% ($[^2P_{1/2}]$ 1/2,3/2)$_1$ − 9.3% ($[^3P_{3/2}]$ 3/2,5/2)$_1$ | 12.11 | 12.116 | 23.82 | 12.160 K |
| Pb XIV | $5p^6 5d^1$ ($^2D_{5/2}$) − $5p^5 5d^2$ | 54.1% ($[^3P_1]$ 3/2,0)$_{3/2}$ − 22.6% ($[^3P_2]$ 3/2,0)$_{3/2}$ | 12.16 | 12.159 | 32.30 | 12.162 C |
| Pb XIV | $5p^6 5d^1$ ($^2D_{5/2}$) − $5p^5 5d^2$ | 68.8% ($[^1G_4]$ 7/2,0)$_{7/2}$ + 12.2% ($[^3F_3]$ 7/2,0)$_{7/2}$ | 12.20 | 12.237 | 64.89 | 12.230 C |
| Pb XIV | $5p^6 5d^1$ ($^2D_{5/2}$) − $5p^5 5d^2$ | 47.5% ($[^3F_3]$ 5/2,0)$_{5/2}$ − 18.9% ($[^3P_2]$ 5/2,0)$_{5/2}$ + 10.5% ($[^1D_2]$ 5/2,0)$_{5/2}$ | 12.25 | 12.243 | 55.98 | 12.249 C |
| Pb XIV | $5p^6 5d^1$ ($^2D_{3/2}$) − $5p^5 5d^2$ | 65.9% ($[^3F_2]$ 5/2,0)$_{5/2}$ − 8.9% $5d^0 5f^1$($[^1S_0]$0,5/2)$_{5/2}$ + 7.9% $_{(}[^1D_2]$5/2,0)$_{5/2}$ | 13.04 | 13.078 | 30.98 | 13.137 C |
| Pb XIII | $5p^6 5d^2$ ($^3F_4$) − $5p^5 5d^3$ | 62.3% ($[^2H_{9/2}]$5,0)$_5$ + 13.2% ($[^2H_{11/2}]$5,0)$_5$ − 10.6% ($[^4F_{9/2}]$5,0)$_5$ | 13.14 | 13.176 | 23.14 | |

Increasing irradiance to $3.2 \times 10^{10}$ W cm$^{-2}$ the Pb emission on the short wavelength side of 12 nm becomes more defined. For the PbSn946 plasma the emission from Sn at 10.34 nm is present but no visible contribution from Sn at 11.35 nm in the alloy spectra is seen. Using the convolved synthetic spectra for each ion stage as an aid it can be seen the emission between 11.1–11.2 nm corresponds to $5p^65d^3 - 5p^55d^36s^1$ transitions in Pb XII, while the feature between 10.6–10.8 nm corresponds to $5p^65d^4 - 5p^55d^46s^1$ transitions in Pb XI, with a small contribution to the feature between 11.6–11.8 nm. The feature at 11.6–11.8 nm corresponds to $5p^65d^5 - 5p^55d^56s$ transitions in Pb X, for wavelengths shorter than 14 nm the Pb spectrum peaks in emission around 12.88 nm corresponding to $5p^65d^N - 5p^55d^{N+1}$ (where N = 3–5) emission from Pb X–Pb XII ions. The falloff in emission at wavelengths longer than 16.5 nm indicates a shift away from ion stages lower than Pb IX with Sn absorption features still present in the alloy plasmas, while the emission from Pb VIII and Pb IX at 17.33 nm involving 5d–5f and 5p–6d arrays stays constant; this feature is observed for higher laser power densities and corresponds to the emission from the outer plasma.

When the focal spot is further reduced to 400 μm the spectra shift slightly back to longer wavelengths and lower ion stage emission for a laser irradiances of $2.9 \times 10^{10}$ W cm$^{-2}$ and $4.2 \times 10^{10}$ W cm$^{-2}$, a point of note here is that when calculating $T_e$ for a given laser irradiance the CR model does not take into account laser plasma coupling. For the alloys at $2.9 \times 10^{10}$ W cm$^{-2}$ and $4.2 \times 10^{10}$ W cm$^{-2}$ the emission from Sn at 10.34 nm has now disappeared while the emission at shorter wavelength is pronounced compared to the Pb plasma. For a laser irradiance of $4.2 \times 10^{10}$ W cm$^{-2}$ a number of Pb peaks form between 12 and 12.5 nm persist up to a power density of $7 \times 10^{10}$ W cm$^{-2}$. At $5.5 \times 10^{10}$ W cm$^{-2}$ five peaks can be identified at 12.11 nm $^1S_0 - (1/2,3/2)_1$, 12.16 nm $^2D_{5/2} - (3/2,0)_{3/2}$, 12.20 nm $^2D_{5/2} - (7/2,0)_{7/2}$, 12.25 nm $^2D_{5/2} - (5/2,0)_{5/2}$ and 13.04 nm $^2D_{3/2} - (5/2,0)_{5/2}$ corresponding to $5p^65d^N - 5p^55d^{N+1}$ transitions of Pb XIV and Pb XV, with the 12.11 nm and 12.16 nm lines remaining fixed for all spectra at higher power densities. At $7 \times 10^{10}$ W cm$^{-2}$ the three prominent peaks at 10.48 nm $^1G_4 - (9/2,1/2)_4$, 10.6 nm $^3F_3 - (9/2,1/2)_4$ and 10.63 nm $^3F_4 - (11/2,1/2)_5$ correspond to $5p^65d^2 - 5p^55d^26s^1$ transitions in Pb XIII are visible.

For the higher power density of $1.1 \times 10^{11}$ W cm$^{-2}$ the plasmas appear slightly cooler and the emission from Sn has its comparative maximum and shifts the spectrum to longer wavelengths for the PbSn6535 plasma. It is also worth noting at this laser irradiance, Sn emission peaks at a slightly longer wavelength for the PbSn6535 plasma, due to a reduction in Sn absorption. For Pb a number of small emission peaks appear around 13 nm, the peaks at 13.04 nm $^2D_{3/2} - (5/2,0)_{5/2}$ and 13.14 nm $^3F_4 - (5,0)_5$ corresponding to the $5p^65d^N - 5p^55d^{N+1}$ transitions in Pb XIV and Pb XIII respectively, and remain visible up to a power density of $1.2 \times 10^{12}$ W cm$^{-2}$.

For these higher laser power densities, there is an obvious contribution from higher ion stages with a 5p ground configuration, however it is not possible to assign a label to any of these lines without ambiguity. Increasing the power density to $1.6 \times 10^{11}$ W cm$^{-2}$ the Pb spectrum now appears relatively flat, a number of lines around 14.8 nm are visible but weak, and with increasing power density the intensity of the transitions decreases and appear as photoabsorption features at the highest power density of $1.2 \times 10^{12}$ W cm$^{-2}$. At $2 \times 10^{11}$ W cm$^{-2}$ and $2.6 \times 10^{11}$ W cm$^{-2}$ a number of lines on the short wavelength side of the peak at 12.16 nm are evident, which is now the longest wavelength peak of the cluster. These peaks again originate from 5p–5d transitions of ion stages Pb XVI to Pb XX. A relatively flat band of emission between 13 and 15 nm is observed with preliminary analysis suggesting a significant contribution from satellite lines in this region, additionally emission at wavelengths longer than 15 nm falls off more sharply.

At the highest power density of $1.2 \times 10^{12}$ W cm$^{-2}$ the PbSn6535 plasma shows a significant reduction in emission at longer wavelengths, with the emission now falling off beyond 13 nm. The pure Sn plasma exhibits a number of lines between 14.3 and 15 nm from ion stages in the high teens, while longer wavelength emission is significantly reduced compared to the plasma produced at $1.1 \times 10^{11}$ W cm$^{-2}$. However, the shorter wavelength features from lower ion stages are still present indicating emission from the outer plasma. For the PbSn6535 plasma relative to the pure Pb plasma

the reduced intensity of the $5p^65d^1 - 5p^55d^2$ line at 12.16 nm indicates a reduction in self-absorption. The Sn emission from the alloy target decreases with increasing concentration indicating a $T_e$ too high for Sn ions with an opened 4d subshell, while the small emission feature from Sn at 13.5 nm in the PbSn946 results from recombining ions in the outer plasma and has an overlap with the $5p - 5d$ subarray emission from recombined Pb ions seen in the pure Pb spectrum. While the PbSn6535 plasma has emission from lower ion stages, it appears to favor Pb emission which is again evident at longer wavelengths where emission from Pb VIII to Pb X is present at 16.5 nm and 17.33 nm. The crest of the emission now spans from 11.15 nm to 12.16 nm for PbSn6535, with peaks that appear in all three Pb plasmas and arise from ion stages ranging between Pb XIV and approximately Pb XXII.

### 4.4. Influence of Spot Size on Plasma Expansion

When comparing spectra from different spot sizes it was found that for a larger focal spot similar spectral flux can be achieved at a lower power density albeit a greater laser energy; with an emission structure that resembles a lower average charge. The effects of spot size on plasma dynamics has been studied by Harilal *et al.* [43] and Tao *et al.* [44], combining their work and applying to the current spectra; when the spot size increases the plasma propagation dynamics change from being spherical in nature to expanding more cylindrically. The smaller focal spot undergoes more lateral expansion but has a reduced plasma scale length and thus less self-absorption explaining the stronger emission at shorter wavelengths. For larger focal spots there is increased coupling between the plasma and laser due to the cylindrical expansion; as the laser energy is now shared between a greater number of emitting ions a lower average charge is obtained but an increase in continuum emission is observed for the same reason, bremsstrahlung emission scales with $\xi_{Av}^2 n_i$, where $n_i$ is the ion density. It is also worth noting the increased scale length for larger focal spots leads to greater self-absorption and emission from lower ion stages at longer wavelengths from recombined ions and free electrons in the outer plasma [45,46].

## 5. Conclusions

Spectra from Sn, Pb and Sn-Pb alloys have been recorded in the EUV spectral region from both Nd:YAG and $CO_2$ LPPs. With increasing power density, the spectra of pure Sn targets are dominated by the 13.5 nm UTA emission. However, in the alloys this feature appears to be quenched by absorption due to Pb 5p–5d transitions, here the overall intensity does indeed increase but the spectra actually appear flatter with less evidence of discrete structure. Further evidence for this behavior is provided by the emission from optically thinner $CO_2$ LPPs where the emission structure reflects the Sn target concentration and ions present. The greater spectral efficiency for the alloys indicates that they should have a useful role as LPP sources for metrology applications. In addition, the contributing ions stages were discussed and a number of transitions in ion stages Pb XII to Pb XV have been identified.

**Author Contributions:** E.S. performed the experimental setup, experiments, calculations, data analysis and writing of original draft; supervision performed by G.O. and F.O.; review and editing performed by G.O., F.O., P.H. and I.T. All authors have read and agreed to the published version of the manuscript.

**Funding:** Science Foundation Ireland: Award 07/IN.1/B1771.

**Conflicts of Interest:** The authors declare no conflict of interest.

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
