# Peer review of "An Investigation of Laser Produced Lead-Tin Alloy Plasmas between 10 and 18 nm"

_atoms, doi:10.3390/atoms8040075_

Round 1
Reviewer 1 Report
The manuscript "An investigation of laser produced Lead-Tin alloy
plasmas between 10 and 18 nm" by Enda Scally et all presents the
results of a systematic study performed on Pb-Sn alloys along with
spectra from pure Pb and Sn. I recommend this article for publication
after very minor corrections.
1. Table 2: Are all terms in Table 2 even? If there are odd terms this should be noted
The same line 5 below the Table 2 and throughout the article
Some typos:
Page 14, line 5: principle --> principal
Page 16, sect 4.3, end of paragraph: Table 2Error! Reference source not found..
This should be corrected
Author Response
Dear Sir,
Thank you for taking the time to read my paper and your feed back.
- Table 2: Are all terms in Table 2 even? If there are odd terms this should be noted
The same line 5 below the Table 2 and throughout the article
I have used a superscript 'o' on the odd parity jj term values in the table and text.
Some typos:
Page 14, line 5: principle --> principal
Page 16, sect 4.3, end of paragraph: Table 2Error! Reference source not found..
This should be corrected
I have corrected these typos on Pg 14 and 16
Also I have increased the size of all the graphs and moved the text around to account for this increase as requested by the second reviewer; along with changing a few typos I noticed, that can be found in word tracker. Please see attached.
Regards
Enda

Reviewer 2 Report
Title: An investigation of laser produced Lead-Tin alloy plasmas between 10 and 18 nm
The manuscript presents systematic study performed on Pb-Sn alloys of concentration 65-35% and 94-6% by weight along with spectra from pure Pb and Sn in the wavelength range of 9.8 - 18 nm. The spectra from Sn, Pb and Sn-Pb alloys have been recorded in the EUV spectral region from both Nd:YAG and CO2 LPPs lasers. The contributing ion stages and line emission have been identified by the authors using the steady state collisional radiative model of Colombant and Tonon, and the Cowan suite of atomic structure codes.
The research presented in the manuscript presents both experimental and theoretical character, which makes the manuscript very valuable. Moreover, the manuscript is very well written and the results are interesting, and presented in clear and accessible way. The paper evidently contributes to the progress in the field and the importance of the research is clearly demonstrated in the manuscript.
In my opinion this paper should be published, but after taken into account the suggestion presented below.
1. The authors should correct the second paragraph on page 5 in the place “…The 4d - 4f transitions first appear as the feature on the far right in Sn IV…” because the first commented ion is Sn V.
2. The figures should be bigger and the authors should change in whole manuscript axis descriptions in order to avoid misleading, i.e. e.g. Wavelength/nm should be changed to Wavelength [nm].
3. It is very difficult to distinguish in figures for example type of transitions in grayscale or Black and White print version.
In conclusion the investigated in the manuscript matter matches the issues presented in Atoms.
Author Response
Dear Sir,
Thank you for taking the time to read my paper and for your feedback.
- The authors should correct the second paragraph on page 5 in the place “…The 4d - 4f transitions first appear as the feature on the far right in Sn IV…” because the first commented ion is Sn V.
I have corrected Sn IV to Sn VII.
2. The figures should be bigger and the authors should change in whole manuscript axis descriptions in order to avoid misleading, i.e. e.g. Wavelength/nm should be changed to Wavelength [nm].
I have increased the size of all the graphs and moved the text around to account for this increase and I have changed all axis descriptions e.g. Wavelength/nm changed to Wavelength [nm].
3. It is very difficult to distinguish in figures for example type of transitions in grayscale or Black and White print version.
I made the addition to figure 4 caption of adding "(colour online)", I could not see any other way around this issue that would not generate too many figures.
I have also added a superscript 'o' on the odd parity jj term values in the table and text and changed some typos:
"Page 14, line 5: principle --> principal
Page 16, sect 4.3, end of paragraph: Table 2Error! Reference source not found.."
as requested by the other reviewer.
Along with this I changed a few typos I noticed, that can be found in word tracker. Please see attached.
Regards
Enda
